A review on the traditional uses, nutritive importance, pharmacognostic features, phytochemicals, and pharmacology of Momordica cymbalaria Hook F

Mohammed Firdous Sayeed firdous.oncology@gmail.com 1
Babu Dinesh 2
Irfan Zainab 3
Fayed Marwa A.A. marwa.fayed@fop.usc.edu.eg 4
1 Department of Pharmacology, Calcutta Institute of Pharmaceutical Technology and Allied Health Sciences , Uluberia, Howrah , West Bengal , India
2 Faculty of Pharmacy and Pharmaceutical Sciences, Katz Group Centre for Pharmacy and Health Research, University of Alberta , Alberta , Canada
3 Department of Pharmaceutical Technology, Brainware University , Kolkata , West Bengal , India
4 Pharmacognosy Department, Faculty of Pharmacy, University of Sadat City , Sadat City , Egypt
Oliveira Sonia
Electronic publication date: 2024 Feb 27
Publication date: 2024
Volume: 12
Electronic Location ID: e16928
Received 2023 Feb 24; Accepted 2024 Jan 22
Copyright: ©2024 Mohammed et al.
Copyright year: 2024
Copyright holder: Mohammed et al.
License: This is an open access article distributed under the terms of the Creative Commons Attribution License, which permits unrestricted use, distribution, reproduction and adaptation in any medium and for any purpose provided that it is properly attributed. For attribution, the original author(s), title, publication source (PeerJ) and either DOI or URL of the article must be cited.
License URL: https://creativecommons.org/licenses/by/4.0/

Keywords: Momordica cymbalaria, Cucurbitaceae, Phytochemicals, Cucurbitacin triterpenoids, Pharmacological activities, Antidiabetic

Funding: The authors received no funding for this work.

==============================
Momordica cymbalaria Hook F. (MC), belonging to the family Cucurbitaceae, is a plant with several biological activities. This detailed, comprehensive review gathers and presents all the information related to the geographical distribution, morphology, therapeutic uses, nutritional values, pharmacognostic characters, phytochemicals, and pharmacological activities of MC. The available literature showed that MC fruits are utilized as a stimulant, tonic, laxative, stomachic, and to combat inflammatory disorders. The fruits are used to treat spleen and liver diseases and are applied in folk medicine to induce abortion and treat diabetes mellitus. The phytochemical screening studies report that MC fruits contain tannins, alkaloids, phenols, proteins, amino acids, vitamin C, carbohydrates, β-carotenes, palmitic acid, oleic acid, stearic acid, α-eleostearic acid, and γ-linolenic acid. The fruits also contain calcium, sodium, iron, potassium, copper, manganese, zinc, and phosphorus. Notably, momordicosides are cucurbitacin triterpenoids reported in the fruits of MC. Diverse pharmacological activities of MC, such as analgesic, anti-inflammatory, antioxidant, hepatoprotective, nephroprotective, antidiabetic, cardioprotective, antidepressant, anticonvulsant, anticancer, antiangiogenic, antifertility, antiulcer, antimicrobial, antidiarrheal and anthelmintic, have been reported by many investigators. M. cymbalaria methanolic extract is safe up to 2,000 mg/kg. Furthermore, no symptoms of toxicity were found. These pharmacological activities are mechanistically interpreted and described in this review. Additionally, the microscopic, powder and physiochemical characteristics of MC tubers are also highlighted. In summary, possesses remarkable medicinal values, which warrant further detailed studies to exploit its potential benefits therapeutically.

Introduction

Medicinal plants are important in contemporary healthcare. There has been a global trend towards increasing the number of plant-based medications. The hunt for potential novel species with various forms of pharmacological activity is necessary. Furthermore, the development of a renewable raw material base for medicinal plant resources is critical. Plant introduction is the first stage of the transition to widespread cultivation, allowing us to analyze the characteristics of plant growth including the formation of naturally occurring chemicals. The research of new prospects of known plant species in illness therapy is critical. As a result, the goal of this article was to examine the implications of presenting Momordica cymbalaria as a little-known species of medicinal plant with a wide variety of biological activities.

Momordica cymbalaria Hook F. is one of the species of the Cucurbitaceae family. The synonyms of this plant are Momordica tuberose (Roxb.) and Luffa tuberosa (Roxb.). This plant is an everlasting herbaceous climber which trails on the ground and climbs on supports with the help of a stem. It is found in India, mainly in Andhra Pradesh, Karnataka, Madhya Pradesh, Maharashtra, and Tamil Nadu, as a weed (Prashanth, Suresh & Maiya, 2013). The plant is grown along bunds or fences and in the fields. The roots of M. cymbalaria are tuberous, which help to maintain perennial or everlasting habits, and are pubescent. Besides, this plant dries up and wanes at the end when the season comes to an end. The roots are 4–8 cm in diameter, light brownish yellow-colored with a typical odor, and extremely bitter. The fractured surface of the root is fibrous. M. cymbalaria plant has a monoecious stem and is very slender. The leaves are orbicular or reniform with a deeply heart-shaped base and the flowers are unisexual. It bears 2–5 flowers in racemes with a pale-yellow corolla and two stamens for each flower, the male flower’s peduncle of M. cymbalaria is 0.05–0.30 cm long, puberulous, filiform, and ebracteate. The female flower is lone on a peduncle of 28 mm in length. The fruits are 20–25 mm long, pyriform with eight sharp ridges, 24 × 15 mm attenuated at the apex, and with the base narrowed into the curved peduncle, which is fleshy, dark green colored, and ridged. The seeds are 4.6 mm long, ovoid-shaped, smooth, and shiny. Flowering appears during October, while the fruits are reaped from November to January. Furthermore, the tender fruits of M. cymbalaria closely resemble the limited variation of bitter gourd, which is utilized as a vegetable in north Karnataka and south Tamil Nadu of India (Parvathi & Kumar, 2002).

Momordica cymbalaria has been studied by many investigators as a medicinal plant possessing various pharmacological and phytochemical properties. Phytochemical studies on M. cymbalaria have revealed that this plant is rich in tannins, alkaloids, amino acids, vitamin C., carbohydrates, fixed oil, and flavonoids (Parvathi & Kumar, 2002; Kameswararao, Kesavulu & Apparao, 2003; Kumar et al., 2010; Kale & Laddha, 2012). Recently, the bioactive principles presented in M. cymbalaria have been utilized as a reducing agent for the synthesis of silver nanoparticles to develop a cost-effective and environmentally acceptable green synthesis approach for silver nanoparticles production (Paulraj et al., 2021). However, despite having diverse ethnomedicinal uses, the therapeutic benefits of M. cymbalaria have not been tested at the clinical level to date. Consolidated information is deficient concerning the present knowledge relevant to M. cymbalaria research. Thus, the current review attempts to consolidate and summarize the scientific data available to date on M. cymbalaria. In this review, the interest is focused on therapeutic uses, nutritional importance, pharmacognostic characteristics, phytochemicals, and pharmacological activities of M. cymbalaria plant. In such a way, the present article describes a simple and comparatively efficient review of M. cymbalaria.

Search strategy

To investigate the published research studies related to the traditional uses, nutritional importance, pharmacognostic features, phytochemicals, and pharmacology of M. cymbalaria, most databases were searched until April 2022: PubMed, Elsevier, Scopus, Google Scholar, and Web of Science were checked to provide up-to-date reported information. The search criteria contained keywords like Momordica cymbalaria, nutritional importance, pharmacognostic characters, phytochemicals, analgesic, anti-inflammatory, antioxidant, hepatoprotective, nephroprotective, antidiabetic, cardioprotective, antidepressant, anticonvulsant, anticancer, antiangiogenic, antifertility, antiulcer, antimicrobial, antidiarrheal and anthelmintic activities. All papers obtained were under consideration and screened to get all data about naturally occurring M. cymbalaria; unpublished results and commercial materials were excluded from this study.

Taxonomic classification

Kingdom: Plantae	
Superdivision: Spermatophyta- Seed plants	
Division: Magnoliophyta- Flowering plants	
Class: Magnoliopsida - Dicotyledons	
Order: Cucurbitales	
Family: Cucurbitaceae - Cucumber family	
Subfamily: Cucurbitoideae	
Tribe: Jolifficae	
Subtribe: Thladianthinae	
Genus: Momordica	
Species: cymbalaria Hook. F.	
Synonyms: Luffa tuberosa (Roxb.), Momordica tuberosa (Roxb.) Cogn. (Global Plants, 2021).	

Morphology of M. cymbalaria

The morphological characteristics of most organs of M. cymbalaria plant, climbing annual or perennial herb (Jeyadevi et al., 2012), are as follows:

– Stem: slender, scandent, branched, striate.

– Leaves: orbicular-reniform in outline, deeply cordate at the base, obtusely lobed with five to seven lobes.

– Fruits: 20–25 mm long, pyriform with 8 sharp ridges, 24 mm × 15 mm attenuated at the apex and with the base narrowed into the curved peduncle, which is fleshy, dark green, and ribbed.

– Seeds: 4.6 mm long, ovoid-shaped, smooth, and shiny.

– Male flowers: peduncle is 5–30 mm long, filiform, puberulous, ebracteate with 2–5 flowers in racemes with a pale-yellow corolla and two stamens for each flower.

– Female flowers: solitary on a peduncle of 28 mm in length.

– Roots: woody, tuberous, and perennial.

Traditional uses

Different parts of M. cymbalaria have been used traditionally for the treatment of several ailments. The fruits are utilized as a stimulant, tonic, laxative, and stomachic. They are used for treating gout, rheumatism, spleen, and liver diseases. The plant is applied in local folk medicine as an abortifacient and to fight against diabetes mellitus. The juice of fruits and tea leaves of M. cymbalaria are used to treat diabetes, malaria, colic, sores, wounds, and infections. The juice is also used against worms and parasites. They are also useful as an emmenagogue, for measles, hepatitis, and fever. The roots of M. cymbalaria possess abortifacient and aphrodisiac activities. The roots are also utilized to treat constipation, indigestion, diabetes, diarrhea, and rheumatism. The juice of the fruits, leaves, and seeds of M. cymbalaria possess anthelmintic properties (Fernandes et al., 2007; Osinubi et al., 2008).

Nutritional values

The nutrient contents of M. cymbalaria are summarized and correlated with the nutritional value of Momordica charantia in Table S1. Momordica charantia, called bitter gourd or bitter melon, is a very popular plant for healing hyperglycemic conditions in the Ayurvedic system of medicine. This plant is a tropical and subtropical vein of Cucurbitaceae. It contains carbohydrates, protein, calcium, potassium, sodium, iron, copper, manganese, zinc, phosphorus, vitamin C, and β-carotenes. Calcium is the most important mineral for the growth of bones and teeth. It also maintains normal cardiac rhythm, blood coagulation, muscle contraction, and nerve responses. Momordica cymbalaria contains a higher amount of calcium than M. charantia. The iron content in both vegetables is almost the same. Potassium, sodium, copper, manganese, and zinc contents are also high in M. cymbalaria, whereas β-carotenes content in M. cymbalaria is very low. The fruits of M. cymbalaria are reported to contain citric acid, malic acid, and vitamin C (Parvathi & Kumar, 2002).

Pharmacognostic studies

Microscopic characters of tubers

The tubers of M. cymbalaria are reported to contain a periderm in the deeper part of the cortex. The peripheral component of the cortex contains deep narrow fissures. Periderm contains phellem and phelloderm. Phellem is 300 µm broad while the phelloderm is 500 µm broad. Phellem has thin-walled tubular cells and the phelloderm has radial files of thin-walled rectangular cells. The inner portion of the periderm encloses parenchymal cells and starch grains. Nests of vascular strands are present in the midpoint of the tuber consisting of one or two broad xylem materials and a collection of small xylem materials (Koneri, Balaraman & Saraswati, 2006). Phloem is present in the peripheral section of the xylem strand.

Powder characters of tubers

The powder of tubers has been stated to contain cork cells, xylem vessels with pitting, and prisms of calcium oxalate crystals of different sizes (Dhanarajan, Abraham & Isaac, 2006).

Physico-chemical constants

The reported percentages of total ash, acid-insoluble ash, water-soluble ash, and sulfated ash of M. cymbalaria tuber are given in Table S2 (Dhasan, Jegadeesan & Kavimani, 2008). The ash values of the tuber of M. cymbalaria suggest that a certain quantity of inorganic foreign matter and resistant constituents such as sand, soil, and stone particles are present in crude drugs.

Phytochemistry

It is essential to investigate the phytoconstituents of medicinal plants to correlate the relationship between the chemical constituents and the associated pharmacological effects of that plant. Many therapeutically active substances are present in M. cymbalaria, i.e., tannins, alkaloids, amino acids, vitamin C, carbohydrates, and β-carotenes. The fruits of M. cymbalaria contain citric acid, maleic acid, and vitamin C (Gopu & Taduri, 2021). The fixed oil present in the fruits of M. cymbalaria also contains palmitic acid, oleic acid, stearic acid, α-eleostearic acid, and γ-linolenic acid (Firdous et al., 2009; Kale & Laddha, 2012; Kameswararao, Kesavulu & Apparao, 2003; Parvathi & Kumar, 2002). Data obtained from the preliminary phytochemical screening of the tubers of M. cymbalaria are specified in Table S3.

GC-MS analysis

Gas chromatography-mass spectroscopy (GC-MS) analysis of 70% ethanolic extract of the tubers of M. cymbalaria revealed the identification of several compounds of different classes as follows (Kumar et al., 2010):

– Aliphatic compounds; Methyl isovalerate, Methyl iso-butyrate, 1-decanol, 2-hexyl, Dichloroacetic acid, 4-hexadecyl ester, Oleic acid, Chloroacetic acid, tetra decyl ester, Ethyl undecenoate, Myristic acid, Stenol, Margaric acid, Pentadecanoic acid, Arachidic acid, 6-Tetradecane sulfonic acid, butyl ester, Pentafluoro-propionic acid, Hepta-decyl ester, and Ethyl linoleate.

– Aromatic compounds, plant acids and esters as; Cyclopentane acetic acid, 2-n-Propylthiane, 2-Furanocarboxyaldehyde,5-hydroxy methyl, 1-Ethyl-2-Pyrrolidinone, Ethanol 2-(3,3-dimethylcyclohexylidene)-(Z)-,1-Isopropenyl-3-Propenyl-Cyclopentane, Bicyclo- [2.2.1 Heptane, 2-(1-Buten-3-yl)-3-Deutero, 2,5,Cyclohexadiene-1,4-dione,3-hydroxy2-methyl-5-(1- methyl ethyl), 1H-Cycloprop[e]azulen-4-ol-decahydro1,1,4,7-Tetramethyl-[1aR-(1aα,4β,4aβ,7α,7aβ,7bα, Ethyl N-(O-anisyl) formimidate, delta.2-tetrazaboroline,1,4,5,triethyl, and Cyclohexanone,5-ethenyl-5-methyl-4 (1-methyl ethenyl)-2-(1-methylethylidene)-c.

– Terpene Derivatives as; Linalool oxide and (-)-isopulegol.

– Phenolic Compounds as; 2- Methoxy-4-vinylphenol.

– Hydrocarbons as; (trans)-2-nonadecene, n-pentatriacontane, and n-octadecane.

– Steroid components as; Androstan-17-one, and 3-ethyl-3-hydroxy-(5 alpha)-.

– Cyclo-oligisilanes as; Hexa-T-butyl cyclotrisilane.

– Pyrimidine base as; Thymin. GC-MS analysis of the methanolic extracts from in vivo grown plants of leaves of M. cymbalaria revealed the identification of the following compounds (Fig. 1) (Gopu et al., 2021): n- Hexadecanoic acid, 9-octadecenoic acid methyl ester, Octadecanoic acid methyl ester, 17-octadecynoic acid, 3,3-Diaminobenzidine, Aspidospermidin-17-ol,1-acetyl-19, 21-epoxy-15,16-dimethoxy, Cholesterol, Cholestanol, E-8-methyl-9-tetradecen-1-ol acetate, Cholestan-3ol-2-methylene, Vitamin E, Octacosanoic acid methyl ester, 9,10-secocholesta-5,7,10(19)-triene-3,24,25-troil, Ethyl iso-allocholate, Spirot-8-en-11-one,3-hydroxy-(3β-5α,14β,20β,22β,25R), 5β-cholestane-3α,7α,12α,24α,25-pentol, Cholestan-3-ol,2-methylene, Stigmasterol, 9,10 secocholesta-5,7,10 (19)-triene-3,24,25-triol, Ethyl iso-allocholate, β-sitosterol, 1-Heptatriacotanol, Ethyl iso-allocholate, Methyl triacontanoate, 9,10-Secocholesta-5,7,10 (19)-triene-3,24,25-triol, Lupeol, 9,10,secocholeste-5,7,10 (19)-triene-3,24,25-triol, Stigmata-3,5-dien-7-one, Tetra-hexadecamethiol, and Lanosta-7,9 (11)-dien-18-oic acid,22,25-epoxy-3,17,20-trihydroxy-y-lactone.

The GC-MS analysis of the bioactive compounds present in the methanolic extracts of in vitro leaf callus derived from in vivo grown plants of M. cymbalaria revealed the identification of the following compounds (Fig. 2) (Gopu et al., 2021): Pyrrolidine,1-nitro, 2-pyrrolidinone, 3- Aminopiperidin-2-one, Diethyl Phthalate, Cyclohexanol,4- [(trimethylsilyl)oxy], cis, 3,7-Dihydroxy-5,6-epoxycholestane, Tricyclo [20.8.0.0(7,16)] triacontane,1(22),7 (16)-diepoxy, Tricyclo [20.8.0.0(7,16)] triacontane,1(22),7(16)-diepoxy, Spirost-8-en-11-one,3-hydroxy(3β,5α,14β,20β,22β,25R), 5-(7a-Isopropenyl-4,5-dimethyl-octahydroinden-4-y)-3-methylpent-2-enal, Octacosanoic acid, methyl ester, Propanoic acid,2-(3-acetoxy-4, 4,14-trimethyl androst-8-en-17-yl), Ethyl iso-alcoholate, 9,10-Secocholesta-5,7,10 (19)-triene-3,24,25-triol (3β,5Z,7E), and 1-Heptatriacotanol.

Figure 1 Some compounds identified by GC-MS analysis from the methanolic extracts of the leaves of M. cymbalaria.

Figure 2 Some compounds identified by GC-MS analysis of the methanolic extracts of in vitro leaf callus derived from in vivo grown plants of Momordica cymbalaria.

The GC-MS analysis of bioactive compounds present in the methanolic extracts of the roots derived from in vivo grown plants of M. cymbalaria revealed the identification of the following compounds (Fig. 3) (Gopu et al., 2021): 1H-pyrrole-2,5,dihydro-1-nitroso, 1,6-Anhydro-2,4-dideoxy-β-D-ribo-hexopyranose, 4-Hydroxy-2-methylacetatophenome, 9-Octadecenal, d-Mannose, Diethyl phthalate, n-Hexadecanoic acid, Triacontanoic acid methyl ester, 5β,7β-H,10α-Eudesm-11-en-1α-ol, Pregnane-3,11,20,21-tetrol, cyclic 20,21-(butyl boronate) (3α,5β,11β,20R), Oxirane,2,2-dimethyl-3- (3,7,12,16,20-penta methyl, 3,7,11,15,19 -heneicosa pentaenyl), Spirost-8-en-11-one,3-hydroxy, (3β,5α,14β,20β,22β,25R), Spirost-8-en-11-one,3-hydroxy, (3β,5α,14β,20β,22β,25R), Spirost-8-en-11-one,3-hydroxy, (3β,5α,14β,20β,22β,25R), Ethyl iso-allocate, Stigmasterol, and β-sitosterol.

Figure 3 Some compounds identified by GC-MS analysis of the methanolic extracts of root derived from in vivo grown plants of Momordica cymbalaria..

Flavonoids

The total flavonoid content and estimation of rutin in the methanolic extract of M. cymbalaria fruits were performed by a spectrophotometric method based on the formation of complexes with aluminum chloride; the extract was reported to contain 0.47% (w/w) and 0.27% (w/w) of total flavonoid content and the amount of rutin, respectively. The existence of rutin was also identified by chemical tests and thin-layer chromatography (Kale & Laddha, 2012).

Hydrocarbons

The long-chain saturated hydrocarbons were segregated from the petroleum ether extract of M. cymbalaria fruits which may serve as a marker component for further characterization and standardization of crude drug and marketed formulations (Kale & Laddha, 2013).

Sterols

The steroidal content was calculated and reported, where the steroid content was found to be maximum in the refluxed methanolic fruit extract (80.38 ± 0.03 sitosterol equivalence mg/gm) followed by hexane (78.74 ± 0.91 sitosterol equivalence mg/gm) and ethyl acetate (57.96 ± 0.10 sitosterol equivalence mg/gm) at 100 mg/ml concentration (Srinivasulu et al., 2017).

Triterpenes

In another study, the fruit powder of M. cymbalaria was subjected to successive solvent extraction, and six compounds were isolated. The isolated compounds were characterized using infrared, mass, 1H-, and 13C-NMR spectral data. Four known cucurbitacin triterpenoids (momordicosides) and a novel compound, 21, 22-didehydroxy momordicoside, were isolated and characterized from the fruits of M. cymbalaria (Dhasan, Jegadeesan & Kavimani, 2008).

The three compounds identified as 3,7,23-trihydroxy-cucurbita-5,24-diene-19-al, 3,7,25-trihydroxy-cucurbita-5,23-diene-19-al and 3,7-dihydroxy-25-methoxy-cucurbita-5,23-diene-19-al, respectively, were reported earlier from Momordica foetida, a perennial climbing vein indigenous to tropical Africa, thoughtfully related to bitter melon. Another two compounds were later identified as momordicoside-A (21,22,23,24-tetrahydroxy-cucurbita-5-ene-3-O-biglucoside) and 23,24-dihydroxy-cucurbita- 5,21-diene-3-O-biglucoside. Besides, quercetin was also isolated and identified (Dhasan, Jegadeesan & Kavimani, 2008).

Mcy, a 17 k Da protein in the aqueous extract of M. cymbalaria fruits with an isoelectric point of 5.0, was identified as an active constituent of antidiabetic action. This protein was perceived to be a novel protein by distinguishing its N-terminal amino acid sequence from those in the protein data bank. A contrast between the N-terminal sequence of the Mcy protein and the α-chain of human insulin was formed since both are antihyperglycemic proteins (Rajasekhar et al., 2010).

Insulin α-chain Gly Ile Val Glu Gln Cys Cys Thr Ser Leu Tyr

Mcy protein Gly Leu Glu Pro Thr Thr Thr

Similarly, such insulin-mimetic peptide was also found in other plant species, namely Canavalia ensiformis, Vigna unguiculata, and Bauhinia variegata (Xavier-Filho et al., 2003).

Pharmacological studies

Toxicology of M. cymbalaria

It was reported that M. cymbalaria methanolic extract is safe up to 2,000 mg/kg. Furthermore, no symptoms of toxicity were found during either the short-term (48-hour) or long-term (14-day) monitoring periods (Mahesh Kumar et al., 2018).

The diverse pharmacological activities of M. cymbalaria have been evaluated by many investigators. Different parts of M. cymbalaria are shown to possess different pharmacological effects on various preclinical models (Fig. 4). The pharmacological studies of M. cymbalaria reported so far are summarized below:

Figure 4 Pharmacological activities of Momordica cymblaria.

Analgesic

The ethanolic extract of M. cymbalaria leaves (250 and 500 mg/kg) has been investigated for its analgesic potential on 0.7% v/v glacial acetic acid-induced writhing and radiant heat tail-flickresponse in Swiss albino mice (Ramanath & Burte, 2012). It was noted that the extract (500 mg/kg) significantly decreased the count of writhing in the acetic acid-induced writhing test and increased the mean reaction time in the tail-flick analgesia model. In the acetic acid-induced writhing model, the nociception involves the release of endogenous substances like histamine, serotonin, bradykinin, prostaglandins, and leukotrienes to stimulate the sensory nerve endings, whereas, in the radiant heat tail-flick model, pain is centrally modulated by the central pain pathway. Several complicated processes, viz., opiate, dopaminergic, and serotonergic pathways, are included in the central pain pathway. Therefore, the analgesic activity of M. cymbalaria leaf extract seems to involve both peripheral mechanisms (inhibition of prostaglandin and leukotriene synthesis) and central mechanisms of pain regulation. Furthermore, flavonoids are known to hinder prostaglandin synthesis and cyclooxygenase-2 expressions (Hämäläinen et al., 2011). Thus, the presence of flavonoids in M. cymbalaria may be accountable for its analgesic activity.

Anthelmintic

The anthelmintic activity of petroleum ether, chloroform, ethanolic, and aqueous extracts of the fruits of M. cymbalaria fruits (20 mg/ml) was studied on Indian adult earthworms (Pheretima posthuma). The results showed that chloroform extract took the least time to cause paralysis and death of earthworms, followed by petroleum ether, methanolic, and aqueous extract (Srinivas et al., 2008). The presence of different phytochemicals like tannins, alkaloids, and flavonoids are liable for anthelmintic activity of M. cymbalaria fruits (da Silva, de Carvalho & Borba, 2008; Wang et al., 2010; Athanasiadou et al., 2001). Tannins have been demonstrated to interfere with coupled oxidative phosphorylation causing the blocking of ATP synthesis in these parasites (Martin, 1997). It also binds to the cuticle body surface of the helminth causing paralysis (Williams et al., 2014). Thus, the presence of tannins may be held accountable for the anthelmintic activity of these extracts.

Anticancer and antiangiogenic

The anticancer activity of methanolic extract of aerial parts of M. cymbalaria (100 and 200 mg/kg) was illustrated by the researchers in Ehrlich ascites carcinoma (EAC) bearing Swiss albino mice. The methanolic extract was reported to show a significant decrease in body weight, packed volume, and viable tumor cell count compared to the mice of the EAC control group. The extract also restored the hematological parameters to normal (Jeevanantham et al., 2011a; Jeevanantham et al., 2011b).

The saponin isolated from M. cymbalaria roots was studied on EAC-induced carcinoma in female Swiss albino mice. Treatment with saponin (175 mg/kg) reduced the total cell count and viable cell count. The saponin also significantly increased the survival time of the mice (Koneri et al., 2014).

The antitumor activity of saponin (100 mg/kg) was also evaluated in dimethyl benz[a]anthracene (DMBA) induced breast cancer in female Wistar rats. The isolated saponin reduced tumor size and growth. It also increased the terminal end buds, terminal ducts, alveolar buds, and lobules. The histological improvement was supported by a decrease in necrosis and hemorrhage along with the reduction of focal desmoplastic reaction in the breast of tumor-bearing rats. Moreover, it also reduced the level of lipid peroxidation (LPO) besides enhancing reduced glutathione (GSH) levels and endogenous antioxidant enzymes viz., superoxide dismutase (SOD) and catalase (CAT) (Kaskurthy, Koneri & Samaddar, 2015).

Recently, the saponins of M. cymbalaria roots were studied in diethyl nitrosamine-induced hepatocellular carcinoma in Wistar rats. Oral administration of saponin (175 mg/kg) reduced serum aspartate aminotransferase (AST), alanine aminotransferase (ALT), alkaline phosphatase (ALP), total bilirubin, cholesterol, and triglyceride levels and increased total protein levels. However, there was a significant improvement in SOD and CAT activity and GSH levels in the liver tissue (Nagarathana et al., 2016).

In-vivo antiangiogenic effectiveness of saponins of M. cymbalaria was assessed in air sac angiogenesis in rats and chick chorioallantoic membrane (CAM) angiogenesis models. The air sac model angiogenesis was induced by the administration of carrageenan in the air pouch in Wistar rats. Treatment with saponins (175 mg/kg) significantly reduced the pouch volume, granulation tissue weight, and carmine dye content. In the CAM angiogenesis model, angiogenesis was induced in fertilized chicken eggs by erythropoietin. The experiment was performed between days 8 and 12 of incubation because the implants made from days 8–10 are strongly angiogenic. Saponin (32 µg) was administered into the eggs on the 12th and 13th day after the administration of erythropoietin (30 units) for 4 days from the 8th to 12th day. The saponins were found to reduce vascular formation (Koneri et al., 2014).

In the angiogenesis process, migration of vascular endothelial cells from parental vessels, invasion through the matrix, proliferation, and formation of capillary tubes occurs (Folkman, 2006). Antiangiogenic chemicals generally reduce angiogenesis through the inhibition of proteases or prevention of phosphorylation of receptors resulting in the interruption of endothelial tube formation (Ferrara, 2004). In angiogenesis models, the saponins of M. cymbalaria displayed an antiangiogenic effect (Fig. 5). This finding provides a new explanation for the antitumor effectiveness of M. cymbalaria roots.

Figure 5 Mechanism of antiangiogenic effects of Saponins of Momordica cymblaria.

Anticonvulsant

A study on the effect of ethanolic extract of M. cymbalaria fruits (250 and 500 mg/kg) on pentylenetetrazole (PTZ) and maximal electric shock (MES)-induced convulsions was carried out in Wistar rats. In the case of PTZ-induced convulsions, treatment with the extract deferred the onset of seizures and effectively reduced the duration of the convulsion. The administration of the extract showed a vital reduction in the duration of tonic-clonic seizures and recovery time in MES-induced convulsions (Vangoori et al., 2013).

Many studies on isolated saponin revealed the anticonvulsant effect via voltage-gated Na+channel blockade (Liu et al., 2001) and shortening of open time or prolonging the close time of Ca 2+channels (Kim, Nah & Rhim, 2008). Saponins modulate gamma-aminobutyric acid (GABAergic) function by potentiating [3H]-muscimol binding to GABAA receptors in rat brains (Kim et al., 2001). In addition, saponin also blocks the N-methyl-D-aspartate (NMDA) receptor-mediated excitatory process in rat hippocampal cells (Kim et al., 2004). On the other hand, flavonoids have been reported for their anticonvulsant activity. The mechanisms involved in the anticonvulsant activity of flavonoids are linked to their effect on GABA and NMDA receptors (Citraro et al., 2016). Hence, all these findings support that the anticonvulsant activity of M. cymbalaria fruits might be attributed to the existence of saponin and flavonoids.

Antidepressant

Only one major animal study was conducted to examine the antidepressant activity of M. cymbalaria fruits. The hydroalcoholic extract of M. cymbalaria fruits (at 200, 400, and 600 mg/kg body weight) was evaluated for its antidepressant effect using two behavioral models. viz., forced swim test and tail suspension test in Swiss albino mice. The duration of immobility was acclaimed in both experimental models. Mice treated with hydroalcoholic extract of M. cymbalaria fruits showed a critical lowering in the duration of immobility in both experimental models compared to the mice of the control group (Daripelli et al., 2011). It has been previously reported that flavonoids obtained from medicinal plants like Hypericum perforatum and Glycyrrhiza uralensis show antidepressant activity (Butterweck et al., 2000; Fan et al., 2012). Furthermore, the addition of rutin to Hypericum perforatum extract caused potentiation of antidepressant activity (Nöldner & Schötz, 2002). Interestingly, M. cymbalaria fruits have been reported to contain rutin (Kale & Laddha, 2012). Thus, the antidepressant activity of M. cymbalaria fruits may be possibly attributed to the presence of rutin in M. cymbalaria. However, further investigation is needed to explore the underlying mechanisms behind the antidepressant potential of M. cymbalaria.

Antidiabetic

The most prominent potential pharmacological effect of M. cymbalaria is its antidiabetic activity. The effect of M. cymbalaria fruit powder (500 mg/kg) was studied on the blood glucose level and other biochemical parameters in alloxan-induced diabetes in Wistar rats. Treatment with the powder vitally reduced the blood glucose level and improved the hepatic glycogen level in the powder-treated diabetic rats. The powder also decreased the serum cholesterol and triglyceride levels in diabetic animals (Rao et al., 1999). The aqueous, ethanolic, and n-hexane fractions of M. cymbalaria fruits were studied in both normal and alloxan-induced diabetes in Wistar rats. The blood glucose levels were estimated at 0, 1, 3, 5, and 7 h after the treatment. The aqueous extract of M. cymbalaria at a dose of 500 mg/kg showed a maximal blood-glucose-lowering effect in diabetic rats, whereas the same dose did not exhibit any hypoglycemic activity in normal rats (Rao, Kesavulu & Apparao, 2001). The type 2 antidiabetic activity of saponins of M. cymbalaria roots was studied in streptozotocin-nicotinamide-induced diabetes in Swiss albino mice. Treatment of type 2 diabetic mice with saponin of M. cymbalaria (175 mg/kg) yielded a considerable reduction in blood glucose, cholesterol, and triglyceride levels with an increase in serum insulin level. Moreover, saponin increased the mass of pancreatic β-cells in diabetic mice (Firdous et al., 2009).

In another study, the effect of oleanane-type triterpenoid saponin isolated from the roots of M. cymbalaria on glucose uptake in isolated diaphragms of both diabetics following streptozotocin administration and non-diabetic Swiss albino mice was evaluated. In both models, the diabetic and non-diabetic mice increased glucose uptake in the diaphragm. An increase in β-cells in pancreatic histology was also observed. Besides, the insulin-releasing activity of isolated oleanane-type triterpenoid saponin was investigated in the rat insulinoma cell line (RIN-5F). The release of insulin was elevated in the presence of saponin (1 mg/ml) from RIN-5F pre-exposed to adrenaline (5 µM) and nifedipine (50 µM) (Koneri et al., 2014).

The glucose uptake activity of oleanane-type triterpenoid saponin isolated from the roots of M. cymbalaria was demonstrated in the L6 cell line (mouse skeletal muscle cell line). The saponin (0.01−0.10 mg/ml) did not show cytotoxicity against the L6 cell line and significantly increased the glucose uptake in a concentration-dependent manner. This finding suggests that oleanane-type triterpenoid saponin isolated from the roots of M. cymbalaria can be very effective in treating insulin resistance (Samaddar, Balwanth & Chandrasekhar, 2015). Besides these studies, the effect of “Mcy protein” isolated from the fruits of M. cymbalaria was studied in streptozotocin-induced diabetes in Wistar rats (Marella et al., 2015). The Mcy protein (2.5 mg/kg) significantly lowered the blood glucose level, serum and tissue lipids, and kidney and liver function markers. This protein also showed pancreatic islet regeneration. Therefore, Myc protein can lessen hyperlipidemia and control diabetes by increasing the regeneration of pancreatic islets.

Increased apoptosis and decreased replication of β-cells reduce insulin secretion (Montanya & Téllez, 2009). It has been well documented that the generation of reactive oxygen species (ROS) in diabetes plays a vital role in β-cells apoptosis (Yang et al. 2011). Phytochemicals such as rutin and quercetin are excellent antioxidants. Some triterpenoid saponins have been reported to exhibit antioxidant effects in rats (Kim et al., 2001). The presence of rutin, quercetin, and triterpenoid saponins has been reported in M. cymbalaria (Kale & Laddha, 2013; Koneri, Balaraman & Saraswati, 2006; Dhasan, Jegadeesan & Kavimani, 2008). Moreover, triterpenoid saponins of M. cymbalaria augmented insulin release in the presence of alpha 2 adrenergic agonists, adrenaline. In a study, the antagonism of alpha 2 adrenergic receptors by yohimbine elevated insulin release and β-cells proliferation (Naghadeh, Moghadam & Ibrahimi, 2006). Hence, there remains a need for further examination of the effectiveness of triterpenoid saponins of M. cymbalaria on alpha 2 adrenergic receptors on β-cells to understand the underlying mechanism of its antidiabetic activity.

Antidiarrheal

The effect of the methanolic extract on the fruits of M. cymbalaria (200, 400, and 600 mg/kg) was evaluated against different experimental models of diarrhea in Wistar rats. Administration of the extract showed a critical inhibitory effect against castor oil-induced diarrhea and prostaglandin E2 (PGE2)-induced entero-pooling in Wistar rats. In the charcoal meal test, the extract displayed a vital decrease in gastrointestinal motility in rats (Vrushabendra Swamy et al., 2008). Interestingly, different reports in the literature revealed that components like flavonoids, saponins, steroids, tannins, and alkaloids are responsible for antidiarrheal activity through diverse mechanisms (Carlo et al., 1994; Macander, 1986). Flavonoids and tannins are recommended to be accountable for antidiarrheal activity by enhancing water and electrolyte reabsorption. Besides these, other compounds exert anti-diarrheal activity by inhibiting intestinal motility (Daswani et al., 2010). Therefore, the antidiarrheal activity of the fruits of M. cymbalaria could be due to the presence of these phytochemicals. Thus, further investigations are needed to establish the antidiarrheal mechanisms of M. cymbalaria.

Anti-inflammatory and antiarthritic

The antiinflammatory activity of methanolic extract of aerial parts of M. cymbalaria (100 and 200 mg/kg) was assessed using carrageenan-induced hind paw edema in Wistar rats. The extract at doses of 100 and 200 mg/kg was reported to produce 53.11% and 44.43% of inhibition of paw edema, respectively, when measured 4 h after carrageenan administration (Jeevanantham et al., 2011a; Jeevanantham et al., 2011b).

In a study on formaldehyde, Freund’s adjuvant, and collagen-induced arthritis, the ethanolic and aqueous extracts of M. cymbalaria fruits were investigated in Wistar rats. Both extracts (200 and 400 mg/kg) significantly reduced paw edema in all three models. Serum aspartate transaminase (AST), alanine aminotransferase (ALT), alkaline phosphatase (ALP), blood urea nitrogen (BUN), creatinine, cholesterol, and triglyceride levels were significantly reduced by the extracts in both Freund’s adjuvant and collagen-induced arthritic rats. A vital increment in total protein and albumin levels was observed. Moreover, both extracts also attenuated the elevated weight of organs like the liver, kidney, and spleen (Reddy, Rao & Sambasiva-Rao, 2015).

However, flavonoids, for example, rutin and quercetin, have been identified in M. cymbalaria (Dhasan, Jegadeesan & Kavimani, 2008; Kale & Laddha, 2012). Rutin and quercetin are well-known naturally occurring flavonoids. Rutin exerts anti-inflammatory activity by decreasing the expression of cyclooxygenase-2 and inducible nitric oxide (NO) synthase (Choi et al., 2014), whereas quercetin inhibited the expression of inflammatory cytokines, cyclooxygenase, and lipoxygenase (Li et al., 2016). Hence, there remains a need for further examination of the outcome of M. cymbalaria on major inflammatory pathways.

Antimicrobial

The cup plate diffusion and minimal inhibitory concentration (MIC) methods were conducted to determine the antimicrobial activity of petroleum ether, chloroform, aqueous and methanolic extracts of M. cymbalaria fruits against different bacteria (Escherichia coli, Staphylococcus aureus, Bacillus subtilis, Shigella sonnei, Klebsiella pneumonia, Salmonella typhi, Proteus vulgaris, and Pseudomonas aeruginosa), and fungi (Candida albicans and Aspergillus niger). The results indicated that the methanolic extract (2 mg/ml) was more effective against all sets of microorganisms (Vrushabendra Swamy & Jayaveera, 2007).

A study involving the agar well diffusion assay on different microorganisms revealed that the ethanolic and chloroform extracts of M. cymbalaria roots show MIC ranging between 1–5 mg/ml against Escherichia coli, Staphylococcus aureus, Bacillus subtilis, Proteus morganii and Salmonella typhimurium. A study on the effect of ethanolic and chloroform extracts on fungi (Candida albicans, Aspergillus niger, Penicillium chrysogenum, Trichophyton rubrum, and Aspergillus flavus) by microtiter plate assay showed MIC ranging between 1–5 mg/ml (Balkhande & Surwase, 2013).

Another study showed that the petroleum ether, chloroform, aqueous and ethanolic extracts of the aerial part of M. cymbalaria display antimicrobial activity against some clinically isolated bacteria viz., Escherichia coli, Staphylococcus aureus, Klebsiella pneumonia, Pseudomonas aeruginosa (clinically isolated) and fungus, Aspergillus niger. The aqueous and ethanolic extracts revealed inhibitory activities on Escherichia coli, Staphylococcus aureus, Klebsiella pneumonia, and Pseudomonas aeruginosa, whereas the chloroform extract showed inhibitory activity on Aspergillus niger (Sajjan et al., 2010).

Several high-quality investigations have demonstrated the relationship between the structures of the compounds obtained from plants and their associated antimicrobial activity, which showed close correlations. Moreover, many research groups have sought to enlighten the antimicrobial mechanisms of natural compounds. Quercetin has been moderately associated with the inhibition of DNA gyrase. It has been proposed that sophoraflavanone G and (-)-epigallocatechin gallate inhibit the functions of the cytoplasmic membrane and licochalcones A and C inhibit energy metabolism (Cushnie & Lamb, 2005). M. cymbalaria contains flavonoids and other phytochemicals, which may represent novel leads, and future investigations may allow the identification of pharmacologically suitable antimicrobial agents from M. cymbalaria.

A recent study concerning the green synthesis of M. cymbalaria extract silver nanoparticles showed an antibacterial effect against multidrug-resistant human pathogens (Gopu et al., 2022).

Antioxidant

The in vitro antioxidant activity screening of the hydroalcoholic extracts of aerial parts, fruits, and roots of M. cymbalaria was carried out by ferric ion reducing, 2, 2′-azino-bis (3-ethylbenzothiazoline-6-sulfonic acid) (ABTS) free radical scavenging, NO scavenging, and total antioxidant assays. The extract of the aerial parts showed greater ferric ion-reducing power than the extracts of fruits and roots. The fruit extract was reported to show higher NO and ABTS radical scavenging activity. The total antioxidant activity of the fruit extract was found to be 95.27 mg equivalent of ascorbic acid per gram (Prashanth, Suresh & Maiya, 2013). The methanolic extract of M. cymbalaria fruits showed 2,2-diphenyl-1-picrylhydrazyl (DPPH) free radical, superoxide anion, hydroxyl radical, and NO scavenging activities. The extract also exhibited inhibition of in-vitro lipid peroxidation (Vrushabendra Swamy & Jayaveera, 2007). Another study on in vitro antioxidant properties of the hydroalcoholic extract of the tubers of M. cymbalaria has shown potent reducing power, superoxide anion, and hydroxyl radical scavenging activities (Pramod et al., 2008). The total phenolic content of the methanolic extract of M. cymbalaria fruits was investigated by the Folin-Coicalteau phenol reagent test. The total phenolic amount was 272.00 ± 2.20 mg gallic acid equivalent per gram of the plant extract. As flavonoids and phenolic compounds are considered good sources of natural antioxidants, the antioxidant activity of M. cymbalaria can be mainly attributed to the presence of flavonoids and phenolic compounds in this plant.

Antiulcer

The anti-ulcer effect of the petroleum ether, chloroform, and methanolic extracts of M. cymbalaria fruits (100 and 200 mg/kg) against aspirin, alcohol (80% ethanol), and pyloric ligation-induced gastric ulcer models in Wistar rats have been reported previously (Kumar et al., 2011). In the aspirin-induced ulcer model, the methanolic extract of M. cymbalaria fruits indicated a significant defensive effect compared to petroleum ether and chloroform extracts. The methanolic extract significantly abridged ulcer index, the volume of gastric secretion, and free and total acidity of gastric secretion in the pyloric ligation-induced gastric ulcer model. All three extracts significantly lowered the ulcer index in the alcohol-induced ulcer model (Dhasan, Jegadeesan & Kavimani, 2010).

In a study on 80% ethanol-induced ulcers in Wistar rats, the aqueous extract of M. cymbalaria fruits (500 mg/kg) lowered total acidity and ulcer index significantly. The extract restored gastric mucosal structure by reducing gastric erosion and lesions. A vital reduction in the gastric lesion and an increase in non-protein sulfhydryl (NP-SH) level and gastric wall mucus concentration were observed in the extract-treated rats (Dhasan, Jegadeesan & Kavimani, 2010).

Flavonoids, the main compounds of medicinal plants, have been demonstrated to have gastroprotective activity, and several gastroprotective mechanisms of flavonoids have been reported so far. Quercetin has anti-secretory and antihistaminic properties, which decrease histamine release from gastric mast cells and inhibit gastric proton pump. Another important mechanism of flavonoids like quercetin and rutin is their antioxidant properties, which involve the scavenging of free radicals, metal ion chelation, inhibition of oxidizing enzymes, and lowering lipid peroxidation. Besides gastroprotective action, flavonoids also accelerate the healing of gastric ulcers (Mota et al., 2009). Interestingly, flavonoids like rutin and quercetin have been identified in M. cymbalaria (Dhasan, Jegadeesan & Kavimani, 2008; Kale & Laddha, 2012). Thus, the existence of these flavonoids may be held accountable for the antiulcer activity of M. cymbalaria.

Cardioprotective

An animal study has been carried out to demonstrate the attenuating effect of the ethanolic extract of M. cymbalaria roots in preventing isoproterenol (ISO)-induced cardiac injury in Wistar rats. Pretreatment with the ethanolic extract of M. cymbalaria roots (250 and 500 mg/kg) prevented the increase of serum lactate dehydrogenase (LDH), creatine kinase (CK), AST, ALT, ALP and attenuated the alterations of oxidative stress markers like LPO, GSH, CAT, and SOD in cardiac tissues (Raju et al., 2008).

Saponins isolated from the M. cymbalaria roots exerted the cardioprotective effect in ischemic reperfusion-induced myocardial injury in Wistar rats and hypoxia-induced cardiomyocyte cell (H9c2) death in vitro. Pretreatment with saponins (25 mg/kg) reduced myocardial damage by improvement in CK and LDH levels. Moreover, saponins reduced the levels of thiobarbituric acid reactive substances (TBARS) in rats, besides increasing the levels of GSH and the activities of SOD and catalase in cardiac tissues. Saponins also significantly recovered the developed tension and heart rate after myocardial damage induced by ischemia-reperfusion in rats. In H9c2 cells, saponin (20 µg/ml) was found to show a protective role against hypoxia-induced cell death (Mulumba, Koneri & Samaddar, 2015).

Both isoproterenol and ischemia-reperfusion increase oxidative stress in cardiac tissue and promote myocardial cell death (Elahi, Kong & Matata, 2009). In ischemic reperfusion injury, increases in intracellular Na+ and Ca2+ resulted in irreversible damage to cardiac tissue (Jennings et al., 1985). Saponins of M. cymbalaria roots have been shown to increase endogenous antioxidants in cardiac tissue (Mulumba, Koneri & Samaddar, 2015), but there remains a need for further examination of the effectiveness of saponins on cardiac physiology.

Diabetic neuropathy

Increased oxidative stress in chronic hyperglycemia is a crucial aspect of the progression of neuropathy. The neuroprotective activity of a triterpenoid saponin isolated from the roots of M. cymbalaria (100 mg/kg) was studied in streptozotocin-induced diabetic male Wistar rats. Neuropathic analgesia was estimated by tail-flick and hot-plate models. It was perceived that triterpenoid saponin significantly decreased tail immersion latency time and increased pain sensitivity in diabetic rats. The histopathological study revealed that there was advancement in the myelination and degenerative modifications of dorsal root ganglion neurons and sciatic nerve fibers. Also, treatment with triterpenoid saponin indicated a significant reduction in LPO levels and increased SOD and catalase activities in the sciatic nerve (Citraro et al., 2016).

The protective effect of saponins isolated from roots of M. cymbalaria has been investigated in high glucose-induced neuropathy in mouse neuroblastoma cells (NB-41A3) neuropathy. The results indicated a vital reduction in aldose reductase activity and the accumulation of sorbitol in NB-41A3 cells on saponins treatment. The saponins also improved the Na+/ K+-ATPase activity and reduced IL-1β, IL-6, and TNF-α production. Moreover, the saponins significantly improved blood glucose levels and lipid profile and decreased glycosylated hemoglobin levels. These results suggest that saponins possess neuroprotective activity in diabetic peripherals (Samaddar, Balwanth & Chandrasekhar, 2015). Moreover, the effect of oleanane-type triterpenoid saponins isolated from the roots of M. cymbalaria was screened in diabetic peripheral neuropathy by in vivo and in vitro methods. In this study, streptozotocin-induced diabetic Wistar rats were employed for various tests of peripheral neuropathy like muscle grip strength, pain sensation test, and nerve conduction velocity measurement. Treatment with oleanane-type triterpenoid saponin (100 mg/kg) increased muscle grip strength, reaction time to pain sensation, and improved nerve conduction velocity. In the in vitro study, saponin significantly decreased the aldose reductase activity and accumulation of sorbitol in sciatic nerve culture (Samaddar, Bhattarai & Chandrasekhar, 2016).

Hyperglycemia increases blood glucose concentration above normal levels and incites glucose-induced neurotoxicity (Tomlinson & Gardiner, 2008). The increase in neuronal glucose level activates the polyol pathway due to the overexpression of aldose reductase, which accumulates sorbitol and promotes neuropathy (Oates, 2002; Tang, Martin & Hwa, 2012). Na+/K+-ATPase plays a critical role in membrane potential. The increase in neuronal glucose concentration reduces Na+/K+-ATPase activity via protein kinase activation (Nagilla & Karnati, 2014). Moreover, the development of advanced glycated end products in hyperglycemia stimulates the production of proinflammatory cytokines (Vlassara et al., 2002). These factors together promote the advancement and promotion of diabetic neuropathy. A triterpenoid saponin isolated from M. cymbalaria roots was found to reduce aldose reductase activity, sorbitol accumulation, improved Na+/K+-ATPase activity, and reduced expression of proinflammatory cytokines (Samaddar, Bhattarai & Chandrasekhar, 2016). Hence, the triterpenoid saponin of M. cymbalaria roots can be attributed to its neuroprotective properties during hyperglycemia.

Effect on fertility

The anti-ovulatory and abortifacient activities of ethanolic extract of M. cymbalaria roots (250 and 500 mg/kg) were studied in female Wistar rats. The extract critically reduced the duration of the estrous cycle and meta-estrous phase and increased the pro-estrous phase, but no change in the diestrus phase was observed. It also possessed a dose-dependent abortifacient activity in pregnant rats during the organogenesis period (Koneri, Balaraman & Saraswati, 2006).

The ethanolic extract of M. cymbalaria roots (250 and 500 mg/kg) was studied at successive stages of embryogenesis in female Wistar rats. The extract displayed highly critical anti-implantation activity. However, an examination of the estrogenic activity of the extract activity did not display any elevation in uterine weight or vaginal cornification. Rats treated with the extract were not found to show any utero-trophic changes, such as the thickness of the endometrium and the height of the endometrial epithelium. However, glucose, cholesterol, and ALP levels in the uterus were not increased compared with the control group (Koneri et al., 2007).

In a study to investigate the progestational and anti-progestational activities, the Sprague Dawley (SD) rats were ovariectomized on the 8th day of pregnancy, and pregnancy was maintained in those rats by administration of estradiol (0.1 µg/rat/day) and progesterone (3 mg/rat/day) for 13 days. The number of viable fetuses on the 20th day was counted, and it was found that the administration of estradiol and ethanolic extracts of M. cymbalaria roots (250 and 500 mg/kg) to the rats did not maintain pregnancy. In the Clauberg assay, the administration of estrogen and ethanolic extract of M. cymbalaria roots (250 and 500 mg/kg) showed ramifications for the uterus. However, the administration of estrogen, norethisterone, and the extract of M. cymbalaria roots did not inhibit the proliferative changes caused by norethisterone (Koneri et al., 2007).

Cucurbitaceae plants have been reported to contain ribosome-inactivating proteins. α- and β-momorcharins are two ribosome-inactivating proteins isolated from the seeds of M. charantia that exhibited abortifacient activity (Ng, Chan & Yeung, 1992). Hence, the abortifacient and anti-implantation effects of the ethanolic extract of M. cymbalaria roots may be because of the existence of the ribosome-inactivating protein. On the contrary, a recent study stated that M. cymbalaria extracts possess a protective effect on diabetes-mediated reproductive toxicity in male Wistar rats (Elangovan et al., 2021). After oral administration of the extracts, the diabetic rats’ reproductive indices, as well as the antioxidant levels of SOD and glutathione-s-transferase (GST), were considerably enhanced (p < 0.05). Besides, the postprandial blood glucose (PBG) and malondialdehyde (MDA) levels were dramatically lowered after the oral administration of M. cymbalaria extracts. It also helped the reproductive organs of diabetic rats to regain their histomorphology. In diabetic rats, peel extract at a dosage of 500 mg/kg was shown to be more effective in raising testosterone levels and sperm count. Accordingly, M. cymbalaria controls not only postprandial blood glucose levels but also improves reproductive health in diabetics (Elangovan et al., 2021).

Hepatoprotective

The hepatoprotective activity of 70% ethanolic extract of tubers of M. cymbalaria was investigated against carbon tetrachloride (CCl4)-induced liver damage in Wistar rats. Pretreatment with 70% ethanolic extract of M. cymbalaria (40 mg/kg) significantly reversed CCl4-induced elevation of serum AST, ALT, ALP, and total bilirubin levels along with the reduction of serum cholesterol and triglyceride levels. The extract also enhanced GSH activity and lowered LPO activity in the liver. This study suggests that the probable functioning of hepatoprotective activity may be related to the antioxidant activity of the extract (Pramod et al., 2008). The in-vivo antioxidant and hepatoprotective activities of 70% ethanolic extract of tubers and fruits of M. cymbalaria (20 and 40 mg/kg) against thioacetamide-induced and sodium fluoride-induced hepatotoxicity in Wistar rats displayed significant inhibition of LPO and elevation of GSH activities. The extracts also decreased the levels of serum AST, ALT, ALP, bilirubin, cholesterol, and triglycerides (Mitta et al., 2021; Pramod et al., 2008). The ethanolic extract of roots of M. cymbalaria (500 mg/kg) has also been studied for hepatoprotective activity against CCl4-induced liver damage in Wistar rats. It was found that the extract elevated the levels of serum AST, ALT, ALP, and total bilirubin. The extract also reduced serum cholesterol and triglyceride levels. In this study, the in-vivo antioxidant activity has been evaluated. The extract reduced LPO, besides improving the level of GSH, CAT, and SOD in rat’s liver (Koneri et al., 2008). In another study, the methanolic extract of M. cymbalaria Hook. (200, 400, and 600 mg/kg) exerted hepatoprotective activity via decreasing the levels of AST, ALT, ALP, and bilirubin, and increasing the level of high-density lipoprotein cholesterol (HDL cholesterol) against CCl4-induced hepatotoxicity in Wistar rats. The extract also decreased liver tissue LPO levels and improved the GSH level. Moreover, histological observations revealed that the pretreatment of the extract protected the animals from CCl4-induced liver damage (Vrushabendra Swamy & Jayaveera, 2007). In all these studies, M. cymbalaria improved the levels of endogenous antioxidants. Hence, the hepatoprotective activity of M. cymbalaria may be because of its rich content of flavonoids, tannins, and vitamin C, which possess antioxidative traits (Kumar et al., 2008).

Nephroprotective

The promising nephroprotective activity of 70% ethanolic extract of tubers of M. cymbalaria (20 and 40 mg/kg) has been evidenced against cisplatin, gentamicin, and paracetamol-induced renal injury in Wistar rats. The extract (40 mg/kg) was reported to reduce urea and creatinine levels in serum and increase body weight in all three models. In the paracetamol-induced nephrotoxicity model, the extract (40 mg/kg) has shown an improvement in GSH level and a decline in LPO activity (Pramod et al., 2011). Nitric oxide (NO) plays a pivotal role in cisplatin, gentamicin, and paracetamol-induced renal damage. The renal nitrative stress manifested by an increase in protein nitration and lipid peroxidation promotes renal damage (Abdelmegeed et al., 2013; Dhanarajan, Abraham & Isaac, 2006; Meng et al., 2017). In this study, the extract (100 µg/ml) showed in vitro NO radical scavenging activity. The preliminary phytochemical analysis indicated the presence of triterpenoids, saponins, and cardiac glycosides in the ethanolic extract of M. cymbalaria tubers (Pramod et al., 2011). Saponins have been reported to show antioxidant activity (Kim et al., 2001). Therefore, the antioxidant activity of saponins may be accountable for the nephroprotective activity of M. cymbalaria tubers.

Wound healing

In-vitro evaluation for the wound healing effect of a transdermal patch prepared from M. cymbalaria tuber extract was reported to enhance wound healing. The development of this extract can act as a good candidate to fasten wound healing, especially in diabetic patients (Saundharya et al., 2022).

Evidence-based justification of traditional uses

Numerous preclinical investigations have been conducted based on numerous traditional applications of M. cymbalaria by various researchers to develop evidence-based uses of this plant. Koneri, Balaraman & Saraswati (2006), for example, established anti-ovulatory, abortifacient, progestational, and anti-progestational properties of M. cymbalaria. The same research group has also shown the antidiabetic efficacy of M. cymbalaria saponin glycoside as an antidiabetic ingredient in vivo and in vitro (Koneri, Samaddar & Ramaiah, 2014). In a preclinical investigation, we discovered that the plant possesses hepatoprotective properties (Kumar et al., 2008), which supports the plant’s traditional usage for liver protection. Furthermore, this herb is traditionally used to cure wounds and infections. Our examination of the literature revealed that M. cymbalaria can heal wounds (Saundharya et al., 2022) and significantly inhibit microbial growth (Balkhande & Surwase, 2013). As a result, the preclinical outcome says that the traditional applications of M. cymbalaria are reasonable based on the researchers’ stated preclinical investigations. However, we discovered from the literature analysis that the diabetes study was more thorough than the other traditional applications of M. cymbalaria. As a result, we believe that there is an obvious need for extensive research to determine the cellular mechanisms of M. cymbalaria in reducing the pathological condition in animal models (preclinical studies), which will give us an idea about the mechanisms of mending the disease condition in humans exposed to M. cymbalaria on a traditional evidence basis.

Discussion

Here, we provide a brief overview of many biological functions as a therapeutic plant of M. cymbalaria. There is a long history of using plants in traditional medicine to treat a variety of illnesses. This has led to numerous scientific studies, some of which have involved animal testing. We discovered a wide range of biological activity, both in vivo and in vitro, from the literature study. Numerous studies have been conducted on pharmacological actions, such as those that are antidiabetic, antifertility, and anticancer. They have proven that M. cymbalaria has hypoglycemia properties in rats with antidiabetic action. Moreover, it has been demonstrated that the insulin-releasing action exists. However, we discovered that no research has been done on the cellular mechanisms of this plant’s antidiabetic action. Studies on this plant’s cardioprotective and anticancer properties also revealed the same result. Most of the time, the focus was on the antioxidant as a mechanism of action; yet, to substantiate the mentioned activities, cell line investigations are required to determine the cellular or molecular mechanisms. Similarly, we could not locate any receptor-based research on the antifertility potential of M. cymbalaria. Furthermore, the separated compounds have not been the subject of in silico docking research, according to the current article, and there has been minimal reporting on the isolated chemicals. Studies using in silico docking are crucial because they provide insight into the likely target that a molecule may interact with. It is interesting to note that we discovered that M. cymbalaria had significant antibacterial action. For this reason, it is important to do a study on how this plant affects different targets involved in protein synthesis or the development of cell walls in bacteria and fungi. All things considered, we can state that many biological activities with M. cymbalaria have been conducted following their traditional usage, yet most of the research lacks cellular mechanisms or cellular target-based investigations.

Conclusion

This review provides a concise overview of M. cymbalaria as a plant of medicinal importance. The use of plants in traditional medicine for various disease conditions has a long history, which has led to a wide range of scientific studies using experimental animals. In this review, a wide range of pharmacological activities are consolidated. It appears that M. cymbalaria has commonly been investigated for its hepatoprotective, antidiabetic, antifertility, anticancer, and antimicrobial activity. The positive results perceived in the published research articles reviewed here are likely due to the presence of multiple bioactive compounds. As M. cymbalaria holds a significant number of bioactive compounds like tannins, flavonoids, and saponins, these may help fight against several diseases. Most of the pharmacological studies of this plant were demonstrated using extracts and saponin fractions. A handful of studies are reported only on isolated oleanane-type triterpenoid saponin. The pharmacological studies using crude extracts have limitations because it is not possible to determine whether the findings are due to a single bioactive compound or synergy between multiple bioactive compounds. Moreover, different extraction processes and the use of different solvents result in variable yields of diverse bioactive compounds that limit our ability to compare the findings between the studies. Such limitations were perceived in the research articles concerning M. cymbalaria. Additionally, there is a lack of robust methodology for proper justification of the pharmacological/toxicological findings and molecular mechanisms reported in the various studies in the literature. Hence, this consolidated review presents a summary of scientific findings that will help the researchers investigate the unrevealed but promising therapeutic findings of M. cymbalaria in the future.

Supplemental Information

Supplemental Information 1 Supplementary Tables

Additional Information and Declarations

Competing Interests

Author Contributions

Data Availability

Marwa Fayed is an Academic Editor for PeerJ.

Firdous Sayeed Mohammed conceived and designed the experiments, performed the experiments, analyzed the data, prepared figures and/or tables, authored or reviewed drafts of the article, and approved the final draft.

Dinesh Babu conceived and designed the experiments, performed the experiments, analyzed the data, prepared figures and/or tables, authored or reviewed drafts of the article, and approved the final draft.

Zainab Irfan conceived and designed the experiments, performed the experiments, analyzed the data, prepared figures and/or tables, authored or reviewed drafts of the article, and approved the final draft.

Marwa A.A. Fayed conceived and designed the experiments, performed the experiments, analyzed the data, prepared figures and/or tables, authored or reviewed drafts of the article, and approved the final draft.

The following information was supplied regarding data availability:

This is a literature review.

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
