# Peer review of "A review on the traditional uses, nutritive importance, pharmacognostic features, phytochemicals, and pharmacology of Momordica cymbalaria Hook F"

_PeerJ, doi:10.7717/peerj.16928_

## Round 0.1 · original submission · Major Revisions

Dear authors,

Please, proceed to revise and edit your manuscript for EXTENSIVE English language revisions.

Copyediting is not provided as a standard publication service. Please ensure the language in this submission is clear and unambiguous, grammatically correct, and conforms to professional standards of courtesy and expression.

This submission deals with a plant that is used in traditional or folk medicine. Please ensure that any claims for therapeutic properties are supported by data or citations of appropriate scientific literature.

Additionally, your manuscript would benefit from schematics depicting the biomedical (or other) utilities of this plant; and/or molecular docking studies. Succinctly, a figure that summarizes the nutritive importance, pharmacognostic features, phytochemicals, and pharmacology of the plant.

Please note that your manuscript was not sent out for peer reviewing at this stage.

---

## Round 0.2 · Major Revisions

Dear authors, thank you for your submission and patience. We have now received enough reviews. Please, address the reviewers' comments, accordingly.

·

Basic reporting

The present era is of functional foods and the comprehensive reviews on traditional and unexplored edible plants may help to improve the understanding of their health benefits and food and nutraceutical activity along with their application in the development of new generation functional foods. The present review " A Review to Focus on The Traditional Uses, Nutritive Importance, Pharmacognostic Features, Phytochemicals, and Pharmacology of Momordica cymbalaria Hook F." is a positive step in that direction and enriches the knowledge of the readers. Overall the review is fine and can be published with the suggested changes after a revision.

Experimental design

The structure of the review article is fine and easy to understand. However, no statistical tool/design is mentioned by the authors to select the papers to include in this review.

Validity of the findings

This a review article and hence its findings are based on previously published studies. A good conclusion is drawn from this study and hence it can help readers understand all the aspects of M. cymbalaria.

Additional comments

please make the following corrections before final acceptance:
1. Check the references. More than one reference styles is used.
2. Line 101: To investigate the published research studies related to the traditional uses: ‘the’ should be non-italic
3. A small paragraph describing the morphology should also be discussed for better understanding of the plant.
4. Line 162: Powder characters of tubers: more information should be added like the quantity of calcium oxalate and the size of calcium oxalate crystals
5. Line 166: The ash values of the tuber of M. cymbalaria suggest that a certain quantity of inorganic foreign matter and resistant constituents such as sand, soil, and stone particles are present in crude drugs. This line is either not required or are present in crude drugs may be replaced with may be present in crude drugs as it is not a fixed thing. And this paragraph should be rewritten as:
The reported percentages of total ash, acid-insoluble ash, water-soluble ash, and sulfated ash of M. cymbalaria tuber are given in Table S2 (P. Bharathi Dhasan, Jegadeesan, & Kavimani, 171 2008). The ash values of the tuber of M. cymbalaria suggest that a certain quantity of inorganic foreign matter and resistant constituents such as sand, soil, and stone particles are present in crude 169 drugs.
6. Line 177: Mention the major carbohydrate with therapeutic activities found in M. cymbalaria.
7. Line 252: The osolated, correct with isolated
8. Line 333: Treatment with saponin (175 mg/kg) , correct: treatment with saponins
9. Line 701: significant number of bioactive compounds like tannin, flavonoid, and saponin. It should be tannins, flavonoids, and saponins. Please check.
10. Line 755: Curr. Sci., check the format.
11. Line 763: Pharm. Mag, check the format.
12. Check references in lines 807, 809, 841, 933,
13. A clarity issue is there with Figure 4
14. Figures 3, 4, and 5 are not mentioned in the text

Reviewer 2 ·

Basic reporting

The manuscript was well reported. However, some improvement is required in discussing the paper's subject matter. Extensive comments on the manuscript are provided as additional comments (4).

Experimental design

The study is well-designed. However, more information is required on the traditional uses and toxicology of Momordica cymbalaria. Extensive comments on the manuscript are provided as additional comments (4).

Validity of the findings

The paper will provide more insight into the pharmacological and nutritional significance of Momordica cymbalaria if the authors address all the concerns raised. Extensive comments on the manuscript are provided as additional comments (4).

Additional comments

The manuscript is a scoping review highlighting the ethnobotany, phytochemistry, nutritive potential and pharmacology of Momordica cymbalaria Hook F.
The review is interesting. However, a major revision is required to improve the manuscript.

Comments

1. There was no critical assessment of the literature in the manuscript. A good review should not only list facts but also provide a deeper interpretation of data in the literature to guide further work, which I think is absent in this manuscript.
2. The ethnobotany section, i.e., traditional uses (Line 127), must be appraised appropriately. The authors should provide a table to summarize how the plant is used for ailments with countries where it is used for particular ailments. To improve this section, the authors should also search for more literature on the traditional uses of Momordica cymbalaria.
3. The authors need to improve the discussion of the phytochemistry section. Isolated phytochemicals from the plants should be thoroughly discussed with their biological activities. The activities elicited by the compounds or reported activities should also be summarized in a Table to highlight the significance of the isolated phytochemicals.
4. The pharmacology section of the manuscript also needs to be improved. The authors should talk about which of the pharmacological activities confirms the folkloric usage of the plants. What are the active extracts? At what doses are the extracts active? Which of the pharmacological activities requires further investigation?
5. After that, the author should provide a table for invivo and invitro studies carried out on the plant. The table should include the assay, extract used/isolated compound, plant parts where the extract or phytochemicals were isolated/detected, assay positive control, results, etc.
6. Lastly, the authors should add a section to discuss the toxicology of the extracts and isolated compounds from Momordica cymbalaria. This information will give insight into whether the plant is therapeutically safe for human consumption.
These published articles can be used as a guide.
https://www.cell.com/action/showPdf?pii=S2405-8440%2823%2907436-4
https://www.sciencedirect.com/science/article/abs/pii/S037887411834707X

Reviewer 3 ·

Basic reporting

The paper “A Review to Focus on The Traditional Uses, Nutritive Importance, Pharmacognostic Features, Phytochemicals, and Pharmacology of Momordica cymbalaria Hook F.‘’ investigate detailed, comprehensive review on the information related to the geographical distribution, morphology, therapeutic uses, nutritional values, pharmacognostic characters, phytochemicals, and pharmacological
activities of Momordica.
The review paper is prepared professionally. It includes a well-crafted abstract and an exhaustive introduction that justifies the research undertaken. The introduction points to the deficiencies in the literature on the subject. The aim is clearly defined. Modern analytical methods were used in the research. The discussion of the results is well prepared. The conclusions are well-defined. The illustrative material is appropriate.
In my opinion, the review paper after corrections, will be suitable for publication in PeerJ.

Detailed comments:
Abstract: Should include some numeric data obtained from the study (if any)
Do not use abbreviations when use first time.
Introduction - The introduction is enough in my opinion. Introduction needs some minor changes.
1-Please add an aintroduction sentence about less known plants. I prepared below ones with some fresh references.
Less knownplants distributed mainly rural areas in the world are gained more popularity in recently. They include high content of non-nutritive, nutritive, and bioactive compounds such as flavonoids, phenolics, anthocyanins, phenolic acids, and as well as nutritive compounds such as sugars, essential oils, carotenoids, vitamins, and minerals. Less known plantshave distinct flavor and taste, excellent medicinal value and health care functions as well
Delialioglu RA, Dumanoglu H, Erdogan V, Dost SE, Kesik A, Kocabas Z (2022). Multidimensional scaling analysis of sensory characteristics and quantitative traits in wild apricots. Tur J Agric For 46 (2):160-172. https://doi.org/10.55730/1300-011X.2968.
Çiçek SS, Pfeifer Barbosa AL, Wenzel-Storjohann A, Segovia JFO, Bezerra RM, Sönnichsen F, Zidorn C, Kanzaki I, Tasdemir D. Chemical and Biological Evaluation of Amazonian Medicinal Plant Vouacapoua americana Aubl. Plants. 2023; 12(1):99. https://doi.org/10.3390/plants12010099
Dawadi, P., Shrestha, R., Mishra, S., Bista, S., Raut, R.K., Joshi, T.P., Bhatt, L.R., 2022. Nutritional value and antioxidant properties of Viburnum mullaha Buch.-Ham. Ex D. Don fruit from central Nepal. Turkish Journal of Agriculture and Forestry, 46 (5), 781-789.

Is it necessary to add taxonomic classification????
Do you have any map for world distribution of this specie???

Can you add more information about traditional use???

Also for wound healing

Experimental design

ok

Validity of the findings

ok

Additional comments

NA

---

## Round 0.3 · Major Revisions

Dear authors, thank you for your resubmission. Sadly, you didn't address critical points such as the lack of debate or discussion from a Critical point of view nor the backups of traditional applications from pharmacognosy and preclinical reports (for example)... I am giving you another opportunity to address these very significant shortfalls. Please, refer to the reviewers comments for more information.

Reviewer 2 ·

Basic reporting

NA

Experimental design

NA

Validity of the findings

NA

Additional comments

The authors did not address my concerns in the last manuscript revision.

Reviewer 3 ·

Basic reporting

Dear Editor,

They made all necessary changes

Experimental design

Dear Editor,

They made most necessary changes. As I suggested before they can start the importance with less know plants. They did not consider this.

Validity of the findings

ok

Additional comments

They made most necessary changes. As I suggested before they can start the importance with less know plants. They did not consider this.

---

## Round 0.4 · Minor Revisions

Dear authors, your manuscript has improved significantly. Please, just revise and proofread for minor details such as spacing, punctuation, improper use of shortening or acronyms; be mindful of taxonomy rules!

Reviewer 2 ·

Basic reporting

The article was adequately reported. However, authors should write the plant's generic name in full at the start of the sentence throughout the manuscript. It is improper to abbreviate a scientific name at the beginning of the sentence.

I suggest that the title be changed to 'A Review on Traditional Uses, Nutritive Importance, Pharmacognostic Features, Phytochemicals, and Pharmacology of Momordica cymbalariaHook

There should be spaces between the generic and specific names in the title.

Experimental design

The study was adequately designed.

Validity of the findings

The review article reported all the necessary findings.

Additional comments

The review article is now ready for publication.

---

## Round 0.5 · Minor Revisions

Dear authors, not sure if i am getting the wrong files or you just uploaded the rebuttal without uploading the properly updated doc file/pdf. The details mentioned before seem in their majority overlooked. See the attached pdf. Additionally, I request checking under the consequence that this is the last time I will request these minor changes. Proofread accordingly. Also, please be consistent in your referencing: some citations have DOI others don't! Include and make sure all the referencing is correct throughout! This is a meticulous work. These comments are also valid for the figures. Thank you. And I hope that the next step is the acceptance of this work.

**Language Note:** The Academic Editor has identified that the English language must be improved. PeerJ can provide language editing services - please contact us at copyediting@peerj.com for pricing (be sure to provide your manuscript number and title). Alternatively, you should make your own arrangements to improve the language quality and provide details in your response letter. – PeerJ Staff

---

## Round 0.6 · accepted · Accept

Dear authors, I can now accept your manuscript for publication. Thank you for sharing your work.